# An *R*-Package for the Deconvolution and Integration of 1D NMR Data: MetaboDecon1D

**DOI:** 10.3390/metabo11070452

**Published:** 2021-07-13

**Authors:** Martina Häckl, Philipp Tauber, Frank Schweda, Helena U. Zacharias, Michael Altenbuchinger, Peter J. Oefner, Wolfram Gronwald

**Affiliations:** 1Institute of Functional Genomics, University of Regensburg, 93053 Regensburg, Germany; Martina.Haeckl@stud.uni-regensburg.de (M.H.); peter.oefner@klinik.uni-regensburg.de (P.J.O.); 2Institute of Physiology, University of Regensburg, 93053 Regensburg, Germany; Philipp.Tauber@vkl.uni-regensburg.de (P.T.); Frank.Schweda@klinik.uni-regensburg.de (F.S.); 3Department of Internal Medicine I, University Medical Center Schleswig-Holstein, Campus Kiel, 24105 Kiel, Germany; h.zacharias@ikmb.uni-kiel.de; 4Institute of Clinical Molecular Biology, Kiel University and University Medical Center Schleswig-Holstein, Campus Kiel, 24105 Kiel, Germany; 5Institute of Medical Bioinformatics, University Medical Center Göttingen, 37077 Göttingen, Germany; michael.altenbuchinger@med.uni-goettingen.de

**Keywords:** NMR, 1D, deconvolution, metabolites, quantification, signal identification

## Abstract

NMR spectroscopy is a widely used method for the detection and quantification of metabolites in complex biological fluids. However, the large number of metabolites present in a biological sample such as urine or plasma leads to considerable signal overlap in one-dimensional NMR spectra, which in turn hampers both signal identification and quantification. As a consequence, we have developed an easy to use *R*-package that allows the fully automated deconvolution of overlapping signals in the underlying Lorentzian line-shapes. We show that precise integral values are computed, which are required to obtain both relative and absolute quantitative information. The algorithm is independent of any knowledge of the corresponding metabolites, which also allows the quantitative description of features of yet unknown identity.

## 1. Introduction

Nuclear magnetic resonance (NMR) spectroscopy is a common method to analyze metabolites in biological fluids such as urine or blood plasma [1,2]. The resulting spectra consist of a large number of signals at different frequency positions (*x*-axis), at which each peak corresponds ideally to a certain metabolite, while its signal volume reflects its concentration in the fluid. However, due to the large number of metabolites present in a typical biological sample substantial signal overlap is commonly observed. Therefore, a precise discrimination between individual overlapping metabolite signals is often not feasible, hampering both signal assignment and accurate quantification. To overcome this drawback several methods have already been developed. One obvious solution is to spread the overlapping signals over more than one dimension [3,4]. However, this comes, in the case of ${}^{1}$H-${}^{13}$C HSQC spectra measured at natural abundance, at the price of reduced sensitivity and prolonged measurement time, which may be partly compensated by the application of non-uniform sampling schemes [5] or by application of so-called ultrafast NMR methods, where the conventional time incrementation has been replaced by spatial encoding [6]. Consequently, for large studies comprising up to several thousand samples or in cases where NMR measurement time is limiting, 1D ${}^{1}$H NMR spectra will be commonly used. Therefore, to obtain accurate quantitative information from 1D spectra containing signal overlap, spectral deconvolution techniques are required. For example, Gaussian or Lorentzian line shapes may be fitted to ${}^{31}$P NMR spectra to analyze single compounds such as inositol phosphates [7]. For the analysis of 1D fluorine spectra of proteins, Hughes et al. developed an approach employing Bayesian information criteria to objectively determine the minimal number of signals required to reproduce the experimental data [8]. In the case of the simultaneous analysis of multiple compounds, one widely used approach is to fit reference spectra of standards to approximate the measured mixture spectrum. This can be done in a semi- or fully-automated fashion using, for example, the commercial software Chenomx (Chenomx Inc. Edmonton, Canada), or freely available tools such as Bayesil [9] or Batman [10]. Common to these methods is their requirement of reference information obtained for pure compounds. Moreover, they do not provide quantitative information for NMR signals of yet unknown identity. A recent approach is the SigMa software [11], which was developed for the analysis of urinary spectra and which is able to obtain quantitative information from both known and unknown metabolites. To this end, SigMa focuses mainly on the analysis of spectral intervals representing only single compounds, while for spectral regions containing signal overlap a binning approach is used.

Spectral deconvolution including a combination of time-frequency analysis and probabilistic sparse matrix factorization has been used as a preprocessing step to reduce noise in NMR spectra [12]. For 2D ${}^{1}$H-${}^{13}$C HSQC spectra, Chylla et al. implemented a deconvolution algorithm based on the fast maximum likelihood reconstruction (FMLR) [13] to provide accurate signal integrals without the use of reference spectra. As an advancement of the previously developed Batman package, a Bayesian deconvolution algorithm for the automated analysis of 2D JRES spectra has been introduced [14]. Additionally, in in vivo NMR analyses spectral deconvolution is an important topic as drastically increased line widths result in a substantial amount of signal overlap. Over the years numerous approaches such as LCModel [15] for metabolite quantitation in in vivo NMR spectra have been developed. An overview of existing approaches was recently given by Barker et al. [16]. Additionally, for high-dimensional metabolomic data generated by means of liquid chromatography coupled-mass spectrometry (LC-MS), fully automated signal deconvolution methods using continuous wavelet transforms were developed [17].

The aim of the present contribution was the development of an easy to use *R*-package for the deconvolution of overlapped signals in 1D NMR spectra without the need for reference spectra. To this end, a fully automated determination of the underlying Lorentzian lines, which is the natural line-form of NMR signals, was implemented. The method is based on previous theoretical work by Koh et al. [18,19], which we adapted to optimally work with highly complex spectra of human biofluids such as urine and plasma. Our approach is also distantly related to work by Schmidt et al. who used Lorentzian line shapes for the deconvolution of electrophoretic NMR data to study non-aqueous electrolytes [20]. The implemented method is tested on 1D NMR spectra of a Latin-square design consisting of defined mixtures of 10 different metabolites commonly found in human biofluids, as well as on real human and mouse urine spectra and human blood plasma spectra. Obtained quantitative results are compared to those of two commonly used commercial software packages, namely the AMIX software v. 3.9.13, May 2012 (https://www.bruker.com, accessed on 2 July 2021) and the Chenomx NMR suite v. 8.6, May 2020 (https://www.chenomx.com, accessed on 1 July 2021).

## 2. Results

The deconvolution approach named MetaboDecon1D, which combines automated peak selection and parameter approximation, was tested employing 10 specimens of a Latin-square design containing 10 geometrically diluted metabolites in each specimen, five mouse urine specimens, five human urine and 20 human EDTA blood plasma specimens. As detailed in the Materials and Methods Section (Section 4), for each specimen a 1D ${}^{1}$H spectrum was acquired using either a 1D NOESY (Latin-square design and mouse urine specimens) or a 1D CPMG (all human specimens) pulse sequence.

### 2.1. Peak Selection Procedure

The peak selection procedure was tested for each of the above mentioned spectra. The threshold parameter δ for the signal-noise-differentiation was manually set to ensure that all clear signals were reliably identified, while only a small amount of noise signals was detected. Figure 1 shows as an example the resulting peak triplets for one real human blood plasma spectrum. The three points $x_{left},x_{middle},x_{right}$ for each peak are marked with a green, red and blue dot, respectively.

For a typical spectrum of mouse urine around 1100 signals were detected in total, in the region on the right-hand side of the water signal between 4.60–0.50 ppm, where almost no signal-free regions are present, this number amounts to 780 signals. In comparison, a typical equidistant bucketing with a bucket width of 0.01 ppm would result in 410 buckets in this region. For human urine typically around 1600 signals were identified in total. For the region on the right-hand side of the water signal this number reduced to around 1000 signals. For human plasma which is a little bit less complex in composition around 430 and 400 signals were detected in total and on the right-hand side of the water signal, respectively.

### 2.2. Results of Parameter Approximation Method

Next, the parameter approximation method was tested for the above mentioned spectra. To this end, for each spectrum the mean squared error (MSE) between the sum of the approximated Lorentz-curves obtained after 10 iterations of parameter optimization and the corresponding experimental spectrum were determined. Please note that prior to computation of MSE values both experimental and approximated spectra were normalized to a total integral of one. The MSE results of all tested spectra are listed in Table 1 (For the Latin-square design only the first five results are shown, the MSE values for spectra 6 to 10 amounted to 1.05 × 10${}^{-9}$, 8.46 × 10${}^{-10}$, 1.64 × 10${}^{-9}$, 6.45 × 10${}^{-10}$, and 7.86 × 10${}^{-10}$, respectively, the same is true for the EDTA plasma data, the MSE values for spectra 6 to 20 amounted to 5.85 × 10${}^{-11}$, 9.53 × 10${}^{-11}$, 5.95 × 10${}^{-11}$, 1.01 × 10${}^{-10}$, 1.81 × 10${}^{-10}$, 1.34 × 10${}^{-10}$, 7.16 × 10${}^{-11}$, 5.07 × 10${}^{-11}$, 4.40 × 10${}^{-11}$, 1.55 × 10${}^{-10}$, 1.46 × 10${}^{-10}$, 1.20 × 10${}^{-10}$, 2.31 × 10${}^{-10}$, 6.86 × 10${}^{-11}$, and 1.16 × 10${}^{-10}$, respectively). In all cases very small MSEs below 2.0 × 10${}^{-9}$ were obtained indicating a reliable reconstruction of the experimental spectra.

As an example, in Figure 2 the sum of reconstructed Lorentz curves after 10 iterations (depicted in red) is compared to the original spectrum (shown in black) for one human plasma spectrum. The MSE amounts to 1.66 × 10${}^{-10}$.

### 2.3. Quantification Results

The correctness of the deconvolution of 1D NMR spectra is inspected through the quantification of the spectra of the Latin-square design, the mouse urine spectra, and the human urine and blood plasma spectra. To this end, the Lorentz curves of selected metabolites are integrated and the concentration is determined with the help of the known reference concentration of TSP (or FA for blood plasma). Please note that for each type of spectrum a small representative set of metabolites was analyzed. For the investigated groups different metabolites are examined:**Latin-square design**: acetic acid, alanine, betaine, citric acid, creatinine, ethanolamine, glycine, histidine, taurine, TMAO**Mouse urine spectra**: 2-oxoglutarate, 3-indoxylsulfate, creatinine, glucose, hippurate, succinate, trimethylamine**Human urine spectra**: alanine, creatinine, glycine, hippurate**Human blood plasma spectra**: alanine, creatinine, glucose, lactate, tyrosine

#### 2.3.1. Latin-Square Design

The actual concentrations of the metabolites of the spectra of the Latin-square design are known. Hence, the concentrations determined by MetaboDecon1D were compared with the real values. Furthermore, the concentrations were also determined with the well-known software AMIX (Bruker), and, additionally, both methods were compared with each other. To obtain absolute concentrations, integral values of the respective metabolites were set in relation to the integral of the TSP reference signal assuming for reasons of simplicity equal *T*_1_ relaxation for all signals. Figure 3 shows for the signal of the methyl group of alanine at 1.48 ppm the correlations obtained for the different comparisons. As shown by the best-fit line and an $R^{2}=0.9991$ in (**a**), the actual concentrations and the values determined by MetaboDecon1D correlate well with each other. This indicates that the deconvolution method reflects the concentration of the metabolite alanine well and that accurate integral values are obtained. Furthermore, the correlation between the actual concentration and the results of the AMIX Software (shown in (**b**)) as well as the correlation between AMIX and the new deconvolution method (shown in (**c**)) are high with $R^{2}$ values above 0.99. Please note that for both AMIX and MetaboDecon1D a slight under-quantification of the actual values was obtained, which is due to differences in the *T*_1_ relaxation of alanine and the TSP reference signal. This under-quantification is easily rectified by a multiplicative correction factor.

Correlation results for the other investigated metabolites are listed in Table 2. Results indicate that both MetaboDecon1D (column **a**) and AMIX (column **b**) precisely determine the concentrations of the investigated metabolites. Furthermore, as shown in column **c**, both methods agree well with each other. Please note that no significant differences were obtained between MetaboDecon1D and AMIX.

#### 2.3.2. Mouse Urine Spectra

So far, MetaboDecon1D was tested on the well resolved spectra of the Latin-square design. The analysis of mouse urine spectra, which contain signals from several hundred compounds in a single spectrum, poses a considerably larger challenge. To this end, the performance of MetaboDecon1D on a set of mouse urine spectra was compared to that of AMIX and Chenomx (Chenomx Inc. Edmonton, AB, Canada), the latter being especially designed to deal with signal overlap. As the true concentrations were unknown, we concentrated on the analysis of correlations between the different methods. For further analyses all values obtained for a given metabolite by one of the different methods were normalized by setting its maximum value to one. The correlation for the metabolite 2-oxoglutarate is shown in Figure 4a for the comparison with AMIX and in Figure 4b for the comparison with Chenomx. With $R^{2}$ values of 0.9701 in (**a**) and 0.9988 in (**b**) both AMIX and Chenomx agree well with MetaboDecon1D. However, MetaboDecon1D agrees especially well with Chenomx as both methods were specifically designed to deal with signal overlap, as is the case for the signals of 2-oxoglutarate at 2.42 and 2.99 ppm (data not shown).

The results of the correlation analyses obtained for the other investigated metabolites are given in Table 3. Data show that MetaboDecon1D agrees well with both AMIX and Chenomx with a consistent albeit not significant better agreement with Chenomx. This indicates that MetaboDecon1D allows for a precise determination of integral values even in spectra of highly complex biofluids such as mouse urine.

#### 2.3.3. Human Urine Spectra

Next, spectra of five human urine specimens were analyzed with MetaboDecon1D, AMIX and Chenomx. As a challenging example the signal of glycine at 3.55 ppm was selected, as it is relatively small and in addition considerably overlapped with other signals. Figure 5a shows the comparison of MetaboDecon1D with AMIX, while Figure 5b depicts the comparison with Chenomx. As for the data of mouse urine all values of each method were normalized by setting for a given metabolite the maximum value of each method to one. For the comparison with AMIX in (**a**) only a moderate $R^{2}=0.7482$ was obtained, which considerably increased to $R^{2}=0.9350$ for the comparison with Chenomx (**b**), clearly showing the importance of proper spectra deconvolution of overlapping signals as achieved by both MetaboDecon1D and Chenomx.

The correlation results for the other investigated metabolites are shown in Table 4. Data show that MetaboDecon1D agrees well with both AMIX and, even better, with Chenomx. This indicates that MetaboDecon1D allows for a precise determination of integral values even in spectra of highly complex biofluids such as human urine.

#### 2.3.4. Human Blood Plasma Spectra

Another major and challenging application of NMR-based metabolomics is the investigation of human blood plasma spectra. Due to the presence of large amounts of human serum albumin, which gives rise to large, broad background signals, samples are either ultra-filtrated prior to measurement, or a CPMG pulse sequence, as performed here, is employed for the suppression of macro-molecular signals. Furthermore, human plasma contains large amounts of glucose causing significant signal overlap in the region from 5.2–3.2 ppm. Therefore, to investigate the performance of MetaboDecon1D under such constraints 20 spectra of human blood plasma were employed in which MetaboDecon1D was compared with both AMIX and Chenomx. The correlation of the metabolite lactate is shown in Figure 6a for the comparison with AMIX and in Figure 6b for the comparison with Chenomx. Prior to analysis data were normalized as described above for urine (Section 2.3.3). Data clearly show that for lactate the agreement of MetaboDecon1D with AMIX $R^{2}=0.9160$ in (**a**) is lower than with Chenomx $R^{2}=0.9706$ in (**b**).

The comparisons for all investigated metabolites are provided in Table 5. Except for lactate, $R^{2}$ values slightly below 1 indicate good agreement between MetaboDecon1D, AMIX and Chenomx. The lower agreement observed for lactate will be discussed in the discussion section (Section 3).

## 3. Discussion

We presented a novel deconvolution algorithm, called MetaboDecon1D, to facilitate automatic deconvolution of complex 1D NMR spectra. The key features of MetaboDecon1D include the curvature-based peak selection approach followed by the deconvolution of overlapping signals in the underlying line shapes. Therefore, these two features will be discussed in more detail in the following. A challenging example is the analysis of the signal of the methyl group of lactate in spectra of human blood plasma as it shows considerable overlap with signals of threonine. Here, for a precise determination of integral values a proper deconvolution in the underlying individual signals is mandatory. For MetaboDecon1D the peak selection procedure and the parameter approximation method for the frequency region of the methyl groups of lactate and threonine in a spectrum of human blood plasma are depicted in Figure 8. The peak selection procedure in (**a**) shows the detected peak triplets in this frequency region. The approximation of the parameters starts in (**b**) with the initial Lorentz curves, which are then optimized in an iterative process until after 10 iterations an optimal agreement between the approximated curves and the experimental spectrum is reached (**c**), as demonstrated by the superposition of all estimated Lorentz curves (**d**). As depicted in (**b**) and (**c**) the deconvolution method detects two Lorentz curves each for both of the doublet signals of lactate and threonine. This clearly demonstrates that MetaboDecon1D allows for a fully automated deconvolution even of highly overlapping signals.

The same frequency region was also analyzed with AMIX as shown in Figure 7a and Chenomx as shown in Figure 7b. As indicated by the small vertical lines in Figure 7a the automated peak detection algorithm of AMIX only detects the doublet signal of lactate and not the smaller underlying threonine signals, which are indicated as shoulders of the lactate signals. In contrast, Chenomx allows for a proper deconvolution of the signals of lactate and threonine. However, a proper deconvolution of highly overlapping signals, as in this example, is only achieved by manual fitting.

Similar results for metabolites showing considerable signal overlap were obtained for 2-oxoglutarate (see Figure 4) and glycine (see Figure 5).

These results show that MetaboDecon1D provides precise signal integrals in a fully automated fashion. As it does not depend on reference spectra, precise integrals can also be obtained for yet unknown compounds. This will allow checking for relevant group differences justifying identification of the corresponding compounds. This is a clear advantage compared to conventional binning approaches where signal overlap is not resolved. In the case that major shifts in signal position occur [21] signal alignment procedures such as icoshift [22] or further developments based on icoshift such as IFFD-icoshift [23] may be used to achieve proper signal alignment, as well as for signals where the assignment to the corresponding metabolites is known relative quantification is straight forward. For absolute quantification of metabolites, signal splittings due to J-couplings have to be considered by summation of the corresponding Lorentz curves, which is easily doable for singlet, doublet and triplet NMR signals.

A limitation of MetaboDecon1D is its inability to resolve two exactly overlapping signals as is the case for the signals of TMAO and betaine at 3.25 ppm (see Table 2). In such cases, approaches such as Chenomx that use reference information of the underlying metabolites have a clear advantage as they also allow the deconvolution of exactly overlapping signals based on additional non-overlapping signals of the metabolites in question.

## 4. Materials and Methods

### 4.1. Datasets

*Latin square design* A Latin-square design consisting of 10 samples was prepared to evaluate the experimental performance of the deconvolution algorithm. For the 10 samples, solutions of acetic acid, alanine, betaine, citric acid, creatinine, ethanolamine, glycine, histidine, taurine and trimethylamine-N-oxide, were geometrically diluted from 5000 to 10 μL, whereby the overall substance concentration in each sample was kept constant.

*Human urine and plasma specimens* The used urine and blood plasma specimens were obtained from participants in the German Chronic Kidney Disease (GCKD) study [24,25], which was executed in accordance with the Declaration of Helsinki and registered in the national registry for clinical studies (DRKS 00003971). All study procedures and protocols were approved by the ethics committees of all participating institutions (Friedrich-Alexander-University Erlangen-Nuremberg, Medical Faculty of the Rheinisch-Westfälische Technische Hochschule Aachen, Charité-University Medicine Berlin, Medical Center-University of Freiburg, Medizinische Hochschule Hannover, Medical Faculty of the University of Heidelberg, Friedrich-Schiller-University Jena, Medical Faculty of the Ludwig-Maximilians-University Munich, Medical Faculty of the University of Würzburg). The study was carried out in accordance with relevant guidelines and regulations. Written declarations of informed consent had been obtained from all study participants before inclusion. All specimens were stored at −80 °C until preparation of NMR samples.

*Mouse urine specimens* Spot urine specimens were collected from male C57BI/6J mice (Charles River Laboratories, Sulzfeld, Germany). All experimental procedures were conducted according to the German law for animal care and were approved by the local authorities (registration number provided by the ethic committee of the Regierung von Unterfranken, 55.2.2.2532.2-1107).

### 4.2. NMR Sample Preparation

For preparation of NMR samples, 400 μL of either one of the 10 metabolite mixtures of the Latin square design or of the human urine or EDTA plasma specimens were mixed with 200 μL of 0.1 M phosphate buffer, pH 7.4, which contained in addition 3.9 mM boric acid to impair the growth of bacteria in the sample and 50 μL of 0.75 (wt) trimethylsilylpropanoic acid (TSP) in deuterium oxide (D${}_{2}$O) as internal reference standard. For both human urine and EDTA plasma in addition 10 μL of 81.97 mmol/L formic acid (FA) were added as a second internal reference standard that is not prone to protein binding. For preparation of the mouse urine specimens, 30 μL of urine were mixed with 370 μL of pure water before addition of the same amounts of buffer and reference substance as described above for the Latin square design. All chemicals were obtained from Sigma-Aldrich, Taufkirchen, Germany.

### 4.3. NMR Measurements

Spectra of human plasma and urine specimens from patients with chronic kidney disease were acquired employing a 1D ${}^{1}$H Carr-Purcell-Meiboom-Gill (CPMG) pulse sequence to achieve effective suppression of macromolecular signals. For the Latin-square design and mouse urine samples a 1D ${}^{1}$H NOESY pulse sequence was employed for optimal suppression of the water signal. All NMR experiments were performed at 298 K on a 600 MHz Bruker Avance III spectrometer (Bruker BioSpin GmbH, Rheinstetten, Germany) using a triple resonance (${}^{1}$H, ${}^{13}$C, ${}^{15}$N, ${}^{2}$H lock) cryogenic probe with *z*-gradients in combination with a Bruker SampleJet sample changer (Bruker BioSpin GmbH, Rheinstetten, Germany). Prior to measurement, thermal equilibration of the samples was permitted for 300 s before automatically locking, tuning, matching and shimming the probe. For all measurements a relaxation delay of 4 s was used. The acquisition time amounted to 3.1 and 2.7 s for the 1D CPMG and NOE spectra, respectively.

### 4.4. Preprocessing

Spectra were semi-automatically Fourier-transformed to 128 k real data points, phase corrected and baseline optimized. Next, prior to spectral deconvolution, the remaining water artifact was removed and all intensities were converted to their absolute values. Furthermore, a 2,5-Mean Smoothing was performed to handle the presence of noise, which otherwise might lead to additional maxima (for more information see Section 4.5.1). In our implementation, each intensity value was replaced by the mean value of itself and its two adjacent neighbors on either side. This procedure was repeated twice.

### 4.5. Deconvolution with Lorentz Curves

The deconvolution was realized by constructing an individual Lorentz curve for each signal: (1)$$y\left(x\right)=A\cdot \frac{\lambda }{\lambda^{2}+{\left(x-x_{0}\right)}^{2}},$$
with the area under the curve A·π, the half width at the half height (HWHH) λ, and the frequency position of the peak maximum $x_{0}$. To compute these parameters for each signal a peak selection procedure and a parameter approximation method are required.

#### 4.5.1. Peak Selection

For the deconvolution of an NMR spectrum, the frequency position of each signal needs to be known. Therefore, an automated curvature-based signal selection procedure was realized as proposed by [18]. To this end, a search for peak triplets consisting of three points per signal is performed. First, the second derivative of the spectrum is computed to obtain the inflection points. If $y^{\prime \prime }\left(x\right)<0$ and $y^{\prime \prime }\left(x+1\right)>y^{\prime \prime }\left(x\right)\leq y^{\prime \prime }\left(x-1\right)$, then *x* depicts a local maximum of the spectrum, i.e., the center point of a peak triplet. Second, the left and right points for each peak triplet are defined by nearest zero crossings, local maxima or plateaus of the second derivative. By this, all signal positions, even of partly overlapped signals will be determined. Figure 8a shows as an example the results of the implemented peak selection procedure for a part of a spectrum of human blood plasma. For the removal of noise, a score is calculated for each peak triplet and if its value is higher than the sum of the mean and δ times the deviation of the signal free region, then the peak depicts a real metabolite signal and is kept for further analysis. Here, δ is a user-set threshold parameter that should be carefully checked by manual inspection. Furthermore, the presence of noise may also cause the occurrence of additional peak triplets on the flanks of real signals. To solve this issue, preprocessing with a 2,5 Mean Smoothing approach was performed (see also Section 4.4).

#### 4.5.2. Parameter Approximation Method

After application of the automatic peak selection procedure the frequency positions of the peak triplets for the metabolite signals are known. Next, the parameters *A*, λ and $x_{0}$ need to be calculated for each signal, with *A*, λ and $x_{0}$ denoting a scaling factor, the half line width at half height of the peak and the position of the peak maximum, respectively. An analytical approach calculates them by solving the linear equation system:(2)$$\begin{array}{c} y\left(x_{left}\right)=A\cdot \frac{\lambda }{\lambda^{2}+{\left(x_{left}-x_{0}\right)}^{2}} \end{array}$$(3)$$\begin{array}{c} y\left(x_{middle}\right)=A\cdot \frac{\lambda }{\lambda^{2}+{\left(x_{middle}-x_{0}\right)}^{2}} \end{array}$$(4)$$\begin{array}{c} y\left(x_{right}\right)=A\cdot \frac{\lambda }{\lambda^{2}+{\left(x_{right}-x_{0}\right)}^{2}} \end{array}$$

The system consists of three Lorentz equations for the three points ($x_{left},x_{middle},x_{right}$) of each peak triplet, which allows the computation of the three unknown parameters of each Lorentz curve (initial Lorentz curves are illustrated in Figure 8b). The resulting formula of the linear equation system was redetermined with Mathematica from Wolfram Research v. 12. (https://www.wolfram.com/mathematica/, 2 December 2020).

As the determination of the initial set of parameters considers each Lorentz curve separately, the superposition of the calculated Lorentz curves gives a spectrum of higher intensity than the original spectrum. Hence, a height adjustment is conducted in an iterative process, which adjusts the intensity values for each point of the peak triplets ($x_{left},x_{middle},x_{right}$) and recalculates the parameters *A*, λ and $x_{0}$ until the minimum deviation between the sum of computed Lorentz curves and the experimental spectrum is obtained as determined by the minimal mean squared error. Individually optimized Lorentz curves after 10 iterations are shown in Figure 8c, while the sum of the optimized curves is illustrated in Figure 8d.

To work reliably with real experimental spectra, the original algorithm [18] required some adjustments such as taking the absolute value of the line-width parameter λ. In addition, to avoid numerical instabilities in the calculation of λ, the positions of the peak triplets are shifted to zero for computation of parameters according to the formulas given in Equation (Equation 5).
(5)$$\begin{array}{c} \begin{array}{c} \delta_{x}=x_{left}\\ x_{left,shift}=x_{left}-\delta_{x}=0\\ x_{middle,shift}=x_{middle}-\delta_{x}\\ x_{right,shift}=x_{right}-\delta_{x} \end{array} \end{array}$$

### 4.6. Quantification Through Integration

The deconvolution of a spectrum should allow for an accurate and precise determination of the individual integrals of the underlying metabolites. To this end, the area under each optimized Lorentz curve is calculated through integration.
(6)$$\int_{0}^{b}y\left(x\right)\;dx=\int_{0}^{b}A\cdot \frac{\lambda }{\lambda^{2}+{\left(x-x_{0}\right)}^{2}}\;dx=A\cdot \lambda \int_{0}^{b}\frac{1}{\lambda^{2}+{\left(x-x_{0}\right)}^{2}}\;dx$$
With the upper integration border b given as the number of data points of the spectrum (i.e., the length of the spectrum) and with the basic integral $\int \frac{1}{a^{2}+x^{2}}\;dx=\frac{1}{a}\cdot \arctan\frac{x}{a}$ and the substitution $u=\frac{x-x_{0}}{\lambda }$ follows:(7)$$\int_{0}^{b}y\left(x\right)\;dx=A\cdot \left(\arctan\frac{b-x_{0}}{\lambda }-\arctan\frac{0-x_{0}}{\lambda }\right)$$

In the simplest case of singlet signals corresponding to a single proton these integrals may be directly converted to concentration values with the help of the known concentration of a given reference signal such as TSP (or FA for blood plasma). Otherwise, multiplet splittings and the number of contributing atoms have to be considered in addition.

### 4.7. Software

MetaboDecon1D was developed in *R* 3.6.1 (The R Foundation for Statistical Computing, 2019). A detailed description of the installation and usage is provided in the Appendix A. Furthermore, the *R*-package of MetaboDecon1D comes with a detailed help function including example data. The R-package MetaboDecon1D for the automated deconvolution of 1D NMR spectra can be downloaded from: https://www.uni-regensburg.de/medicine/functional-genomics/staff/prof-wolfram-gronwald/software/index.html. For a typical 1D spectrum of human urine the computation time is approximately 3 min on a standard PC.

## 5. Conclusions

In conclusion, with MetaboDecon1D we provide an *R*-package for the reliable fully automated deconvolution of 1D NMR spectra that should be generally applicable. Compared to existing approaches it is especially advantageous in cases where no reference spectra are available and/or in cases of strong signal overlap as was demonstrated on several examples.

## Figures and Tables

**Figure 1 metabolites-11-00452-f001:**
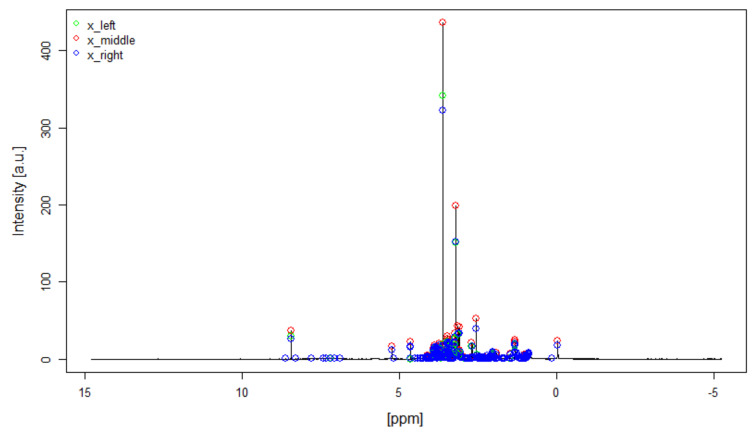
The result of the peak selection procedure is shown for one real human blood plasma spectrum. The detected peak triplets are marked with the green, red and blue dot for the points $x_{left},x_{middle},x_{right}$, respectively.

**Figure 2 metabolites-11-00452-f002:**
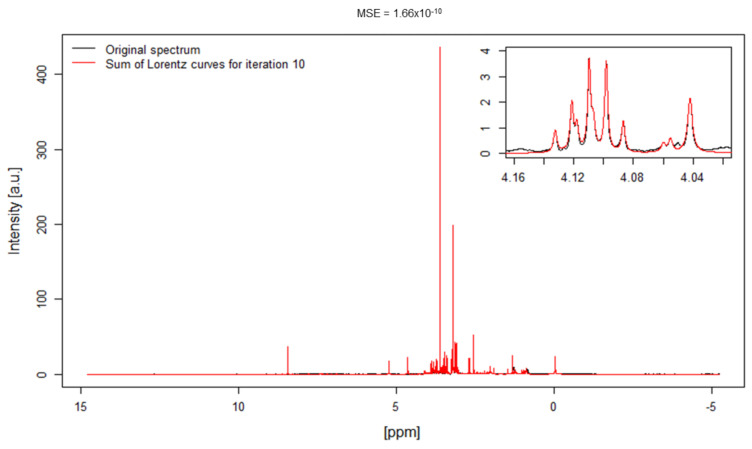
The parameter approximation method is shown for one real human blood plasma spectrum. The original spectrum is represented with the black line and the superposition of all Lorentz curves after 10 iterations is illustrated with the red line. The corresponding MSE value is shown on top of the figure. The insert shows a magnification of a part of the sugar region to show in detail the agreement between experimental and approximated spectrum.

**Figure 3 metabolites-11-00452-f003:**
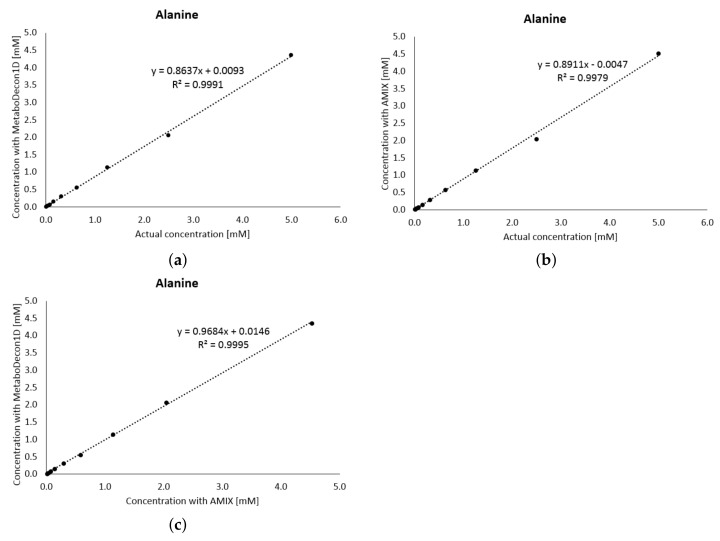
The known actual concentrations of the metabolite alanine of the 10 spectra of the Latin-square design are compared with results obtained by (**a**) MetaboDecon1D and (**b**) AMIX. Furthermore, the correlation between the AMIX and MetaboDecon1D results is shown (**c**). For quantification of alanine by both MetaboDecon1D and AMIX the doublet signal of the methyl group was used.

**Figure 4 metabolites-11-00452-f004:**
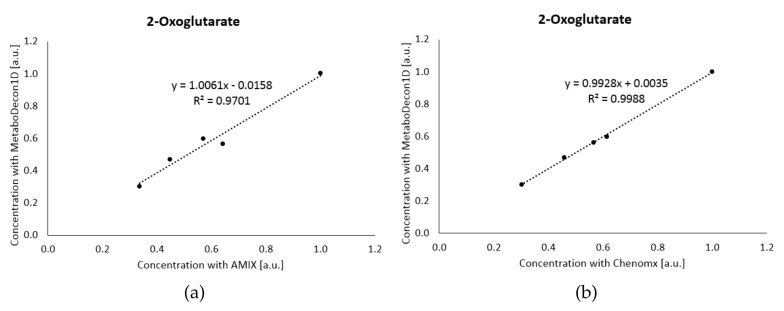
The results of MetaboDecon1D for the metabolite 2-oxoglutarate of the five mouse urine spectra are compared with (**a**) AMIX and (**b**) Chenomx. The best-fit line and the correlation parameter $R^{2}$ are shown.

**Figure 5 metabolites-11-00452-f005:**
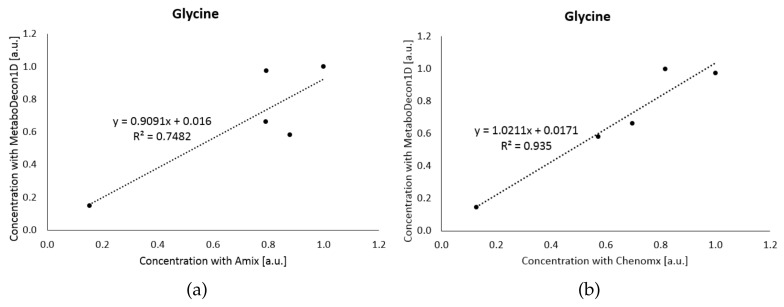
Analysis of glycine in five human urine spectra. The results of MetaboDecon1D were compared with (**a**) AMIX and (**b**) Chenomx. The best-fit line and the fit parameter $R^{2}$ are given.

**Figure 6 metabolites-11-00452-f006:**
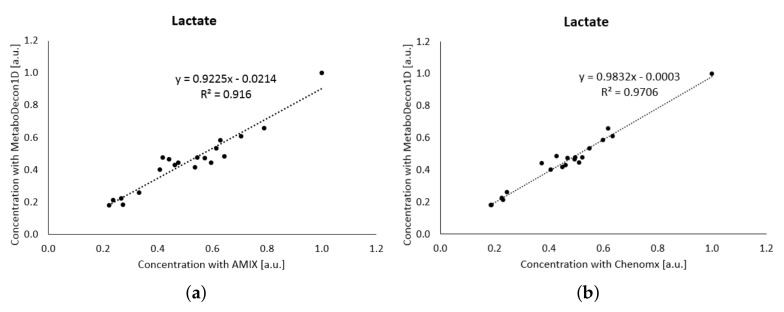
Analysis of lactate in 20 spectra of human plasma. First MetaboDecon1D was compared with AMIX (**a**) and then with Chenomx (**b**).

**Figure 7 metabolites-11-00452-f007:**
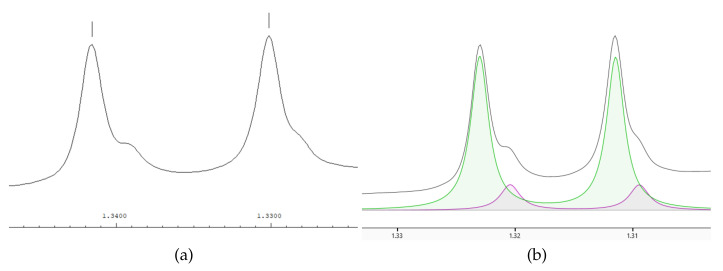
The signals of the methyl groups of lactate and threonine. Automated signal detection by AMIX (**a**) and signal deconvolution by Chenomx for lactate (green area) and threonine (purple area) by manual fitting (**b**).

**Figure 8 metabolites-11-00452-f008:**
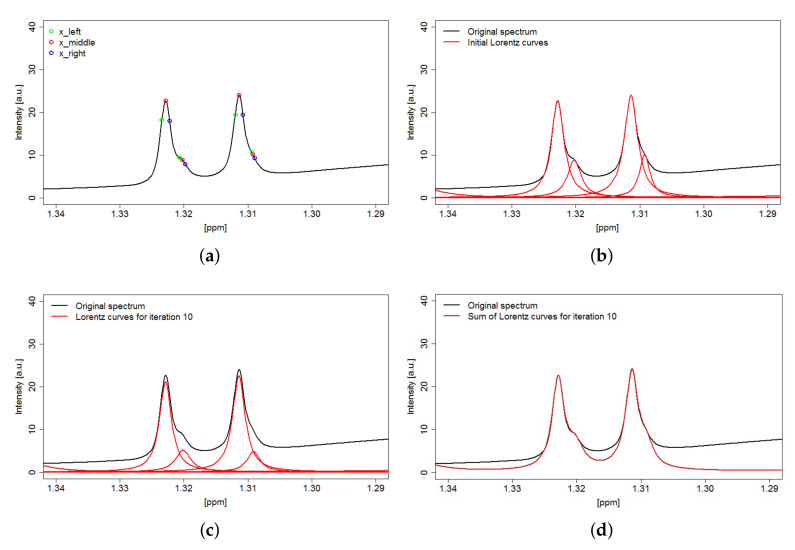
Deconvolution of the frequency region of the overlapping methyl groups of lactate and threonine in a 1D ${}^{1}$H spectrum of human blood plasma. (**a**) Peak selection procedure to find the metabolite positions, (**b**) determination of initial Lorentz curves for parameter approximation, (**c**) iterative approximation of Lorentz curves, showing results after 10 iterations and (**d**) superposition of final Lorentz curves to match experimental spectrum.

**Table 1 metabolites-11-00452-t001:** MSE values after 10 iterations for the first five spectra of the Latin-square design, the five mouse urine, the five human urine and the first five human EDTA blood plasma spectra.

No.	Latin-Square Design	Mouse Urine	Human Urine	Human Blood Plasma
1	6.08 × 10${}^{-10}$	4.41 × 10${}^{-11}$	5.37 × 10${}^{-11}$	1.37 × 10${}^{-10}$
2	6.52 × 10${}^{-10}$	3.58 × 10${}^{-11}$	4.06 × 10${}^{-10}$	1.37 × 10${}^{-10}$
3	7.72 × 10${}^{-10}$	3.35 × 10${}^{-11}$	7.08 × 10${}^{-11}$	1.66 × 10${}^{-10}$
4	8.54 × 10${}^{-10}$	2.92 × 10${}^{-11}$	2.86 × 10${}^{-11}$	8.82 × 10${}^{-11}$
5	7.52 × 10${}^{-10}$	3.11 × 10${}^{-11}$	3.92 × 10${}^{-11}$	1.09 × 10${}^{-10}$

**Table 2 metabolites-11-00452-t002:** Correlation parameter $R^{2}$ for the investigated metabolites for the comparison between the known actual concentrations of the 10 spectra of the Latin-square design and the concentration values obtained by either MetaboDecon1D (**a**) or AMIX (**b**) are shown. Furthermore, the correlations between AMIX and MetaboDecon1D are given (**c**). ${}^{a}$ For the lowest concentration value of ethanolamine at 0.010 mmol/L signals were below the noise threshold. ${}^{b}$ For the three lowest concentration values of taurine at 0.010, 0.020 and 0.039 mmol/L signals were below the noise threshold. ${}^{c}$ For TMAO no values could be determined as one signal of betaine exactly overlaps with the singlet signal of TMAO at 3.25 ppm.

Metabolites	(a)	(b)	(c)	(Selected Signals)
acetic acid	0.9991	0.9992	0.9998	1.91 ppm
alanine	0.9991	0.9979	0.9995	1.48 ppm
betaine	0.9994	0.9969	0.9973	3.25; 3.89 ppm
citric acid	0.9998	1.0000	0.9999	2.56; 2.65 ppm
creatinine	0.9994	1.0000	0.9992	3.05; 4.05 ppm
ethanolamine ${}^{a}$	0.9999	1.0000	0.9999	3.81 ppm
glycine	0.9989	1.0000	0.9989	3.56 ppm
histidine	0.9993	0.9997	0.9985	3.12; 3.22; 3.97; 7.05; 7.77 ppm
taurine ${}^{b}$	0.9996	0.9989	0.9976	3.41 ppm
TMAO ${}^{c}$	-	-	-	3.25 ppm

**Table 3 metabolites-11-00452-t003:** Correlation parameter $R^{2}$ for the metabolites investigated in five mouse urine spectra for the comparison of MetaboDecon1D with AMIX (**a**) and with Chenomx (**b**).

Metabolites	(a)	(b)	(Selected Signals)
2-oxoglutarate	0.9701	0.9988	2.42; 2.99 ppm
3-indoxylsulfate	0.9837	0.9984	7.35 ppm
creatinine	0.9467	0.9907	3.05; 4.05 ppm
glucose	0.9830	0.9776	5.22 ppm
hippurate	0.9985	0.9898	7.54; 7.63; 7.83 ppm
succinate	0.9906	0.9943	2.39 ppm
trimethylamine	0.9949	0.9953	2.87 ppm

**Table 4 metabolites-11-00452-t004:** Correlation parameter $R^{2}$ for the metabolites investigated in the five human urine spectra for the comparison of MetaboDecon1D with AMIX (**a**) and with Chenomx (**b**).

Metabolites	(a)	(b)	(Selected Signals)
alanine	0.9501	0.9955	1.48 ppm
creatinine	0.9990	0.9993	4.05 ppm
glycine	0.7482	0.9350	3.56 ppm
hippurate	0.9997	0.9900	7.54; 7.63; 7.83 ppm

**Table 5 metabolites-11-00452-t005:** Correlation parameter $R^{2}$ for the metabolites investigated in the 20 human blood plasma spectra for the comparison of MetaboDecon1D with AMIX (**a**) and with Chenomx (**b**).

Metabolites	(a)	(b)	(Selected Signals)
alanine	0.9845	0.9715	1.47 ppm
creatinine	0.9802	0.9663	4.05 ppm
glucose	0.9886	0.9615	5.23 ppm
lactate	0.9160	0.9706	1.32 ppm
tyrosine	0.9730	0.9588	6.88 ppm

## Data Availability

Data is contained within the article and Appendix A.

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
