# Peer review of "An R-Package for the Deconvolution and Integration of 1D NMR Data: MetaboDecon1D"

_metabolites, 2021, doi:10.3390/metabo11070452_

Round 1

Reviewer 1 Report

The problem of deconvolution of the 1D spectrum is an interesting and important problem in metabolomic studies. This task can be done with several computer programs. The R software bundle is widely used for a lot of various fields bonded with data analysis. So, the specific package operated under R is an interesting application that can find many users in laboratories and research groups focused on metabolomic studies. The presented manuscript demonstrates high-quality metabolomic experimental data collected for the test and ‘real’ samples obtained from various sources. Unfortunately, we found several drawbacks which have to be addressed in the manuscript:

- the R-package Decon1D practically not describe in the manuscript. Authors presenting an analysis of the collected experimental data, obtained results are compared with results evaluated by two other programs available on the market. Nevertheless, the Decon1D R-package not discussed in the text – no screenshots presented, installation procedure, data preparation, etc.

- I have no idea about the available software for application in research projects. It’s not clear if this software is available for users in repositories like CRAN, Bioconductor, Github, etc, or available from the authors under request. At least we do not find the R-package Decon1D is available for installation in the CRAN deposition source (we have Linux version R 3.6.2 installed on computers). Search with Google provides the software with a similar name written in python (Hughes et al., (2015) PLoS One, 10, e0134474).

Reviewer 2 Report

GENERAL COMMENTS

The paper from Häckl and colleagues describes a new R-package for NMR data deconvolution and integration. The manuscript presents the way the package works and some results on real data, though it lacks more in-depth information that would be helpful to the reader. In fact, actual information on how the package works (inputs, outputs, etc.) is missing, and should at least be reported in the supplementary information. Moreover, the discussion needs some improvement. With these integrations, the paper would be quite interesting for readers in the NMR metabolomics field.

SPECIFIC COMMENTS

Introduction

Line 26: comma after limiting.

Line 27: comma after overlap.

Results

A depiction of the actual output of the R package would be helpful to the reader, to understand exactly what comes out from using the package and how to employ the results. For example: how are signals then identified/named? The reported link for the package does not actually show it, so it was not possible to check this while reviewing the manuscript. Thus an actual look at the way the package works, possibly in the supplementary material would be particularly useful.

Moreover, the package is tested on small dataset, information on how it would work on larger sets (typical for metabolomics), including also time needed for computation, would benefit the readers (and thus, possible package users). This recent paper https://doi.org/10.1016/j.aca.2020.02.025 does similar work and has more detailed information on the software. It might also be cited to refer the pros and cons of your package compared to this.

Discussion

The start of the discussion is a little abrupt. A better introduction to what is going to the results that are going to be discussed is preferrable.

The curves in figure 1a also show a broad singlet signal. A little discussion on this issue would benefit the reader.

Reviewer 3 Report

Comments to Manuscript by Häckl et al.,  

“An R-package for the Deconvolution and Integration of 1D NMR data: Decon1D” 

General comment 

The authors developed a method for perform the automatic deconvolution and integration of 1D NMR spectra, that is one of the major problems in metabolomic studies with a large number of samples. An innovation of the method developed by the authors is the fact that is not necessary a previous knowledge of the compounds present or a spectra database for the spectra deconvolution and integration.

The method proposed seems interesting and could solve problems in the field. However, it should be possible for the reviewers to test the R-package, the package is not in the link indicate in the manuscript. To perform the manuscript evaluation and the novelty of the methodology proposed it is important to test. The authors show the strengths and weakness of the methodology in the results section, that is well constructed, in my opinion in some cases the figures could be improved.  

In my opinion the manuscript is interesting but should be improved for publication on Metabolites.

Comments

  1. The authors do not indicate to which field the spectra were acquired.
  2. In line 24, the authors mention the use of Non-Uniform Sampling for reduce 2D acquisition time, there are other approaches to achieved that. Some of them developed for metabolomic studies, like Ultra-fast NMR. Should be add the references.
  3. In figure 1, it is explicit a drawback of the automatic deconvolution without a spectra image of the compound. The fit of the lactate doublet seems fine, but in the case of threonine is not good, the two peaks of the doublet should be with the same area, but the fit obtained with Decon1D, the two peaks present different areas, roughly more than the double. There is a way to overcome that kind of problems.
  4. In figure 2, should be shown a highlight of a spectra region, to understand better how is perform the peak picking. It will be interesting that the X axis values be the spectra ppm and not an arbitral unit.
  5. Why didn´t the authors used compounds with a long T1 relaxation times in the Latin-square design samples. In my opinion, it will important observe the Decon1D performance in that compounds.
